# Worldwide Prevalence of Epstein–Barr Virus in Patients with Burkitt Lymphoma: A Systematic Review and Meta-Analysis

**DOI:** 10.3390/diagnostics13122068

**Published:** 2023-06-15

**Authors:** Mutaz Jamal Al-Khreisat, Nor Hayati Ismail, Abedelmalek Tabnjh, Faezahtul Arbaeyah Hussain, Abdul Aziz Mohamed Yusoff, Muhammad Farid Johan, Md Asiful Islam

**Affiliations:** 1Department of Haematology, School of Medical Sciences, Universiti Sains Malaysia, Kubang Kerian 16150, Kelantan, Malaysia; 2Department of Applied Dental Sciences, Faculty of Applied Medical Sciences, Jordan University of Science and Technology, Irbid 22110, Jordan; 3Department of Pathology, School of Medical Sciences, Universiti Sains Malaysia, Kubang Kerian 16150, Kelantan, Malaysia; 4Department of Neurosciences, School of Medical Sciences, Universiti Sains Malaysia, Kubang Kerian 16150, Kelantan, Malaysia; 5WHO Collaborating Centre for Global Women’s Health, Institute of Metabolism and Systems Research, College of Medical and Dental Sciences, University of Birmingham, Birmingham B15 2TT, UK

**Keywords:** Burkitt lymphoma, Epstein–Barr, meta-analysis

## Abstract

Burkitt lymphoma (BL) is a form of B-cell malignancy that progresses aggressively and is most often seen in children. While Epstein–Barr virus (EBV) is a double-stranded DNA virus that has been linked to a variety of cancers, it can transform B lymphocytes into immortalized cells, as shown in BL. Therefore, the estimated prevalence of EBV in a population may assist in the prediction of whether this population has a high risk of increased BL cases. This systematic review and meta-analysis aimed to estimate the prevalence of Epstein–Barr virus in patients with Burkitt lymphoma. Using the appropriate keywords, four electronic databases were searched. The quality of the included studies was assessed using the Joanna Briggs Institute’s critical appraisal tool. The results were reported as percentages with a 95% confidence interval using a random-effects model (CI). PROSPERO was used to register the protocol (CRD42022372293), and 135 studies were included. The prevalence of Epstein–Barr virus in patients with Burkitt lymphoma was 57.5% (95% CI: 51.5 to 63.4, *n* = 4837). The sensitivity analyses demonstrated consistent results, and 65.2% of studies were of high quality. Egger’s test revealed that there was a significant publication bias. EBV was found in a significantly high proportion of BL patients (more than 50% of BL patients). This study recommends EBV testing as an alternative for predictions and the assessment of the clinical disease status of BL.

## 1. Introduction

Epstein–Barr virus (EBV) is a pathogenic double-stranded DNA human herpes virus 4 (HHV4). It was first discovered as a human-associated virus by Michael Anthony Epstein and Yvonne Barr in 1964 [1]. The virus consists of a 170–180 kb liner of double-stranded (ds) enveloped DNA with a toroid-shaped protein core, a nucleocapsid with 162 capsomers, and external virus-encoded glycoprotein spikes on the surface of the viral tegument [2]. The EBV genome encodes more than 85 genes, which are involved in the pathogenesis of infection and initiating EBV-associated human disease. There are two major types of EBV: type 1 EBV, which is found worldwide, and type 2 EBV, which is mainly detected in Africa [3]. EBV is the most frequent cause of infectious mononucleosis, with primary infections commonly occurring asymptomatically in teenagers and young adults, especially college students, while in adults, the symptoms are more severe. After primary infection, EBV establishes latent and lytic programs [4,5]. During the latent form of infection, the virus persists in the host cells, while during the lytic phase of infection, new infectious virions are produced [1]. Individuals infected with EBV control the virus’s infectious behavior through cytotoxic immune cell reactions mediated by natural killer (NK) cells and CD8+ T lymphocytes [6,7]. Only a few infected individuals develop chronic EBV-associated pathologies, often due to immune deficiencies, genetic predisposition, and environmental factors [8]. Chronic EBV infections are mainly in the epithelial and lymphocytic cells, which have been associated with malignant diseases [1,9]. EBV is very common in the general population; however, only a minority of infected people experience EBV-related pathologies, suggesting that additional risk factors, such as immune deficiencies, genetic predisposition, and environmental factors, are also crucial in the development of these pathologies [10,11,12]. EBV-associated malignancies express different EBV latent gene products, which are involved in the anti-apoptotic functions of B cells and interfere with innate and adaptive immunity, allowing infected cells to escape immune surveillance. Burkitt lymphoma (BL) is a highly aggressive B-cell non-Hodgkin’s lymphoma that is characterized by the translocation and dysregulation of the proto-oncogene MYC as well as hypermutated immunoglobulin gene sequences [13]. BL is derived from germinal center B cells [14]. Histologically, BL demonstrates sheets of monomorphic medium-sized B cells with basophilic cytoplasm, numerous mitoses, and frequent apoptotic bodies. Macrophages are scattered among tumor cells, giving BL a distinctive histologic appearance called the starry sky pattern. Tumor cells express membrane immunoglobulin (Ig) M, Ig light chain, B-cellular antigen, B-cell lymphoma (BCL) protein 6, and a cluster of differentiation (CD) 10, 19, 20, and 22, while showing negative findings for CD 5, 23, and BCL 2 [15,16,17]. The EBV status of tumors affects the expression of the Epstein–Barr virus (EBV)/C3d receptor and CD21. In essence, all cases of endemic BL are EBV-positive and express CD21, whereas the majority of non-endemic BL among patients who are non-immunosuppressed are EBV-negative and do not express CD21 [18].

The initial BL case was reported in the early 20th century. Denis Burkitt observed widespread childhood tumors in Uganda, which were characterized by malignant growths in the jaw and within the abdominal cavity [19,20]. The World Health Organization (WHO) classified three clinical variants of BL based on cancer epidemiology: endemic, sporadic, and immunodeficiency-associated. These variants are histologically identical and have similar clinical behavior [21]. Endemic BL (eBL) presents in the jaw in younger children and abdominally in older children in malaria-endemic regions, predominantly in sub-Saharan Africa and Papua New Guinea. eBL has a 2:1 male-to-female ratio and a median age of 6 years [22,23].

eBL is mainly localized to geographical areas where Plasmodium falciparum malaria is holoendemic. Chronic B-cell activation or promotion of EBV’s oncogenic potential in the presence of malarial co-infection has been postulated to increase oncogenesis [24,25,26]. Sporadic BL (sBL) is distributed worldwide, with the majority of cases occurring in the United States and Western Europe. sBL is more frequent in children, accounting for 20% to 30% of lymphomas in this age group. Adults with sporadic BL are uncommon, accounting for less than 1% of NHL cases in the United States [27]. BL presents within the abdominal region, lymph nodes, and can also be extranodal. The third variant is HIV-associated BL (ID-BL), which is diagnosed at the early stage of HIV infection and prior to CD4+ T-cell decreases [28].

EBV varies in detection among the three clinical variants of BL. Most endemic BLs are associated with EBV, which suggests that the virus has a direct role in lymphoma pathogenesis. About 95% of eBL detect EBV [28], whereas only about 10–30% of EBV is detected in sBL [21], and 20–40% of EBV positives are detected in ID-BL [28].

EBV plays a critical role in the onset of multiple sclerosis, according to growing data from several study fields. It has been proposed that multiple sclerosis (MS) depends on the early immune response to EBV infection because the severity of the EBV primary infection is strongly associated with the onset of MS many years later. The inability to control this infection might result in the colonization of resident memory B-cell and T-cell follicles in CNS-accessible regions, such as tertiary lymphoid structures, which are particularly prone to triggering immunological disease in the CNS. The period of infection is probably a factor in the immune system’s elimination of the viruses, autoreactive T cells, and antibodies that are directed against CNS components [29,30].

EBV-related malignancies are linked to a latent form of infection, in which EBV expresses a limited set of proteins called EBV transcription programs (ETPs) in every tumor cell, including six nuclear antigens (EBNAs), three latent membrane proteins (LMPs), and untranslated RNA called EBV-encoded small RNA (EBERs), which can mediate cellular transformation [31]. EBV infects primary B cells and induces them to proliferate, by expressing viral genes that were identified as EBNA1, EBNA2, ENBA3A, EBNA3C, and LMP1, which are involved in the latency phase of EBV infection [1]. Additional genes that are included in the transforming B cells are LMP2, viral miRNAs, the small non-coding *RNA EBER*, *BZLF1*, and *BRLF1* [32].

The three latency programs that EBV can display are either Latency I, Latency II, or Latency III. A specific, limited set of viral proteins and RNAs are produced by each latency program (Table 1) [33,34].

Studies showed that BL expresses high levels of MYC, and more than 90% show the translocation of the *MYC* oncogene (8q24) onto the immunoglobulin heavy chain (IgH) (14q34). The chromosomal breakpoints of both MYC and IgH vary between sBL and eBL, giving rise to different aetiologic drivers [35]. A translocation of the MYC gene on chromosome 8, including genetic material from chromosomes 2, 14, or 22, is the classic etiology of BL. The majority of translocations (around 80%) involve the Ig heavy chain on chromosome 14, t(8;14), whereas 15% involve the kappa light chain on chromosome 2, t(2;8), and 5% involve the lambda light chain on chromosome 22 [36,37].

EBV-associated malignancies are diagnosed primarily by a biopsy of the primary tumor, with an EBER in situ hybridization test to confirm the presence of EBV [38]. However, due to the difficulty in obtaining a sample of the tumor or poor patient condition, performikng a biopsy might be challenging [39].

Many studies of EBV-associated lymphoma reveal that EBV-DNA may be found in the plasma of most patients with EBV-related malignancies [40]. DNA from EBV-associated lymphoma is derived as naked DNA fragments from apoptotic or necrotic tumor cells [35,38], whereas it is undetectable in non-EBV-associated tumors or healthy people [24]. Although plasma EBV DNA has recently become more important in the diagnosis and management of EBV-associated cancers [41], particularly Hodgkin’s lymphoma (HL) [41,42] and nasopharyngeal carcinoma [43,44], there are limited data on the diagnostic and prognostic significance of plasma EBV DNA for BL. In order to identify EBV in various types of samples, methods such as the heterophile antibody test, immunofluorescence assays, enzyme immunoassays, Western blot, and polymerase chain reaction (PCR) are used. The use of PCR to determine the EBV viral load is becoming more popular in the diagnosis of EBV-related diseases [45].

Artificial intelligence (AI) is now advancing quickly, and its application in medicine is becoming more relevant. To predict or classify based on input data, AI integrates computer science and databases. Machine learning and deep learning are two types of AI used in the medical field to evaluate medical data and acquire an understanding of the pathogenesis of diseases. Recently, an AI application used for EBV has been developed, such as a deep-learning-based EBV prediction method from H&E-stained whole-slide images (WSI) in gastric cancer [46], and deep-learning-based classifiers to detect microsatellite instability and EBV status directly from hematoxylin-and-eosin-stained histological slides [47]. In BL, artificial neural networks and various types of machine learning were used to analyze the gene expression and protein levels by immunohistochemistry of several hematological neoplasia and pan-cancer series in order to predict patients’ survival and the disease subtype classification with a high accuracy [48]. There is no systematic review and meta-analysis of the prevalence of EBV in patients with BL that we are aware of. As a result, the goal of this systematic review and meta-analysis was to determine the prevalence of EBV in patients with BL, which helps in predicting whether populations are at high risk of increasing the number of BL cases corresponding to EBV infection.

## 2. Materials and Methods

### 2.1. Reporting Guidelines and Protocol Registration

This systematic review and meta-analysis were carried out according to the Preferred Reporting Items for Systematic Reviews and Meta-Analyses (PRISMA) [49] and Meta-analysis of Observational Studies in Epidemiology (MOOSE) [50] guidelines. This study protocol (PROSPERO: CRD42022372293) was submitted to the International Prospective Registry of Systematic Reviews database at the University of York, York, UK.

### 2.2. Eligibility Criteria

The study looked for published studies on the prevalence of Epstein–Barr virus among Burkitt lymphoma patients. The screening was carried out to find possible studies that looked at the presence of EBV in Burkitt lymphoma patients without any restrictions.

### 2.3. Literature Search

In total, 3981 studies were retrieved from four electronic databases: PubMed, Scopus, Web of Science, and Google Scholar. The most recent search was in January 2021, for studies on the prevalence of Epstein–Barr viruses among Burkitt lymphoma patients. Burkitt, Burkitt’s, African Lymphoma, Epstein–Barr, EBV, Human Herpesvirus 4, HHV4, HHV-4, and EB virus were used in the search utilizing a combination of Boolean logical operators (‘AND’ & ‘OR’) and the ‘Advanced’ and ‘Expert’ search options (Appendix A). To ensure a thorough method, the references of the included papers were also examined. To organize and filter out duplicate studies, EndNote X9 software was used.

### 2.4. Study Selection

Two authors (M.J.A.-K. and N.H.I.) independently screened the research title and abstract, followed by the entire text, of all studies retrieved from the literature search to determine the matched studies to be included. Excluded studies include review articles, case studies, non-human studies, views, and viewpoints. Data from news accounts and press releases and information acquired from blogs and databases were not considered. With M.F.J., F.A.H, A.A.M.Y., A.T., and M.A.I, disagreements regarding inclusion were discussed and a consensus was reached.

### 2.5. Data Extraction

The data from the included studies were accessed independently by two authors (M.J.A.-K. and N.H.I.). Before the data extraction procedure, all non-English language studies were translated into English using Google Translate. The data extracted from each of the eligible studies was imported into a predetermined Excel spreadsheet. The following are the extracted data from the selected studies: author name, study type, country, number of BL patients, participants’ age, number of EBV positives in BL, sample type, and EBV detection method. Any discrepancies, or confusing or unfounded data were discussed among the authors in order to reach an agreement. If the problem remains, the corresponding or first author of each study was emailed for clarification.

### 2.6. Quality Assessment and Publication Bias

The quality of the included studies was assessed using Joanna Briggs Institute’s critical appraisal tools. The studies were defined as poor-quality (high risk of bias), moderate-quality (moderate risk of bias), or high-quality (low risk of bias) if the overall score was ≤49%, 50–69%, or ≥70%, respectively [51,52]. Egger’s test was used to verify the funnel plot’s asymmetry. To evaluate publication bias, a funnel plot was constructed to compare the prevalence estimate against the standard error.

### 2.7. Data Analyses

To address the inconsistency among the included studies, a tau-squared test was used to assess heterogeneity (I^2^), where *p* < 0.05 was regarded as statistically significant. A greater homogeneity was regarded as an I^2^ value close to zero, where I^2^ values between 25–50% indicated low heterogeneity, 51–75% indicated moderate heterogeneity, and >75% indicated significant heterogeneity. Based on the critical assessment tools, two authors M.J.A.-K. and N.H.I.) evaluated the quality of each of the included studies by using the critical assessment tools.

Sensitivity analyses and Galbraith plots were also used to assess the quality of the results and identify potential causes of heterogeneity, respectively. The following strategies were used to conduct sensitivity analyses: excluding small studies (*n* < 100); excluding low-quality studies (high risk of bias); excluding studies that did not disclose the prevalence of EBV in patients with BL; only considering cross-sectional studies; and excluding outlier studies. All analyses and plots were generated by using RevMan software (version 5.3.5), RStudio (version 1.1.463), and the metafor package (version 2.0-0) of R software (version 3.5.1) [53].

### 2.8. Subgroup and Sensitivity Analyses

For subgroup analysis, the prevalence of EBV in patients with BL was analyzed through four-time interval trends (1969–1982, 1983–1995, 1996–2008, and 2009–2021); methods of EBV detection (nucleic acid hybridization, polymerase chain reaction (PCR), immunofluorescence, in situ hybridization (ISH), ISH+PCR, and southern blot); and geographical locations (Sub-Saharan Africa, Northern Africa, Southern America, Southern Asia, Northern America, Europe, Eastern Asia, and South-eastern Asia). The studies were categorized based on the sociodemographic index (SDI). To measure social and economic development, the SDI, which ranges from zero to one, employs data on the world’s economies, educational systems, and fertility rates. The SDI is divided into five categories: high SDI (lower bound to upper bound: 0.805129 to 1), high–middle SDI (lower bound to upper bound: 0.689504 to 0.805129), middle SDI (lower bound to upper bound: 0.607679 to 0.689504), low–middle SDI (lower bound to upper bound: 0.454743 to 0.607679), and low SDI (lower bound to upper bound: 0 to 0.454743) [54]. To identify the source of heterogeneity and check the robustness of the results, sensitivity analyses were performed using the following strategies: (1) excluding small studies (<100); (2) excluding low-quality studies (high risk of bias); (3) considering only cross-sectional studies; (4) considering only case-control studies; (5) considering only cohort studies; (6) considering only studies where the age was less than 18 years old; and (7) excluding the outlier studies.

## 3. Results

### 3.1. Selection and Inclusion of Studies

From the database search, 3981 studies qualified for initial screening, and then 2130 studies were excluded due to being duplicate studies (*n* = 1778), review articles (*n* = 259), case reports (*n* = 86), and non-human studies (*n* = 7). Therefore, 1851 studies were further assessed for eligibility by a detailed screening of the titles, abstracts, and full text. Finally, after excluding 1716 studies because they did not comply with the objective of this study, 135 studies were eligible to be included in this systematic review and meta-analysis, as illustrated in the PRISMA flow diagram (Figure 1).

### 3.2. Study Characteristics

Our literature search yielded 135 studies [37,38,39,40,41,42,43,44,45,46,47,48,49,50,51,52,53,54,55,56,57,58,59,60,61,62,63,64,65,66,67,68,69,70,71,72,73,74,75,76,77,78,79,80,81,82,83,84,85,86,87,88,89,90,91,92,93,94,95,96,97,98,99,100,101,102,103,104,105,106,107,108,109,110,111,112,113,114,115,116,117,118,119,120,121,122,123,124,125,126,127,128,129,130,131,132,133,134,135,136,137,138,139,140,141,142,143,144,145,146,147,148,149,150,151,152,153,154,155,156,157,158,159,160,161,162,163,164,165,166,167,168,169,170,171] published between 1969 and 2021, which examined the prevalence of EBV in patients with BL. Detailed characteristics and references of the included studies are presented in Table 2. Overall, this meta-analysis reports data from 4837 patients with BL lymphoma (34.7% female). The ages of these patients ranged from 2.1 ± 2.5 to 47.7 ± 31.8 years (mean ± SD; range, 0.7–98.0). The studies came from eight different regions, and these region groupings were based on the geographic regions defined under the Standard Country or Area Codes for Statistical Use (known as M49) of the United Nations Statistics Division [55]: region unspecified (*n* = 414), Sub-Saharan Africa (*n* = 2104) [56,57,58,59,60,61,62,63,64,65,66,67,68,69,70,71,72,73,74,75,76,77,78,79,80,81,82,83,84,85,86,87,88,89,90], Northern Africa (*n* = 507) [91,92,93,94,95,96,97,98,99,100,101,102,103,104], Southern America (*n* = 801) [105,106,107,108,109,110,111,112,113,114,115,116,117,118,119,120,121,122], Southern Asia (*n* = 37) [123,124,125], Northern America (*n* = 201) [126,127,128,129,130,131,132,133,134,135,136,137], Europe (*n* = 296) [138,139,140,141,142,143,144,145,146,147,148,149,150,151,152,153,154,155], Eastern Asia (*n* = 437) [156,157,158,159,160,161,162,163,164,165,166,167,168,169,170,171], and South-eastern Asia (*n* = 40) [172,173,174,175]. Multiple techniques were used to investigate the presence of EBV in patients with BL, including the use of single and combined methods of nucleic acid hybridization [61,63,73,79,80,81,133,134,160], polymerase chain reaction (PCR) [57,69,85,87,92,98,101,102,109,115,125,126,127,131,139,150,176], immunofluorescence [62,65,66,67,70,71,75,76,86,95,96,130,135,153,177,178,179,180], immunoassay [58,64,74,77,138,148,170], in situ hybridization (ISH) [60,68,72,78,82,83,88,89,90,91,93,97,99,104,105,106,108,112,116,117,118,119,120,121,124,128,129,132,140,141,142,143,144,146,147,149,151,152,154,155,156,157,158,159,161,162,163,164,166,168,169,171,173,174,181,182,183,184], Southern blot [111,136,145,165,167,185,186], and ISH+PCR [103,107,113,114,123,137,172,187,188]. The included studies were conducted between 1969 and 2021, and these studies were divided into four time groups with a fixed interval of 13 years for each: the groups of studies were from 1969 to 1982 [61,62,65,66,67,70,71,73,76,79,80,81,84,86,96,130,133,134,153,177,178], from 1983 to 1995 [57,63,78,83,93,94,95,100,101,109,111,112,122,127,128,129,136,139,141,143,144,145,146,148,151,154,155,156,165,167,170,176,179,180,183,185,186], from 1996 to 2008 [58,60,72,75,85,87,90,91,99,102,103,105,106,107,108,110,113,114,115,120,121,124,126,131,135,137,140,147,150,152,157,159,160,161,163,164,166,172,173,174,182,187,188], and from 2009 to 2021 [56,59,64,68,69,74,77,82,88,89,92,97,98,104,116,117,118,119,123,125,132,138,142,149,158,162,168,169,171,175,181,184,189,190].

Studies were categorized based on the socio-demographic index (SDI) into five categories: high SDI [126,127,128,129,130,131,132,133,134,135,136,137,139,142,143,144,145,146,147,148,151,153,154,155,157,158,159,161,163,165,166,167,170,171,176], high–middle SDI [91,94,95,96,99,100,102,103,107,109,112,116,138,140,141,149,150,152,173,174,175], middle SDI [92,93,97,104,105,106,108,110,113,114,115,117,118,119,120,121,122,123,156,160,162,164,168,169,172], low–middle SDI [59,62,68,74,82,83,87,125], and low SDI [56,57,58,60,61,64,66,67,69,70,71,72,75,76,77,78,79,80,81,84,85,86,88,89,90,98,101,124].

### 3.3. Outcomes

The pooled prevalence of EBV in patients with BL was 59.4% (95% CI, 54.1–64.6%, *n* = 4837), as illustrated in Figure 2.

### 3.4. Subgroup Analyses

Based on the subgroup analyses of the prevalence of EBV in patients with BL over four time intervals, we found a gradually decreasing prevalence of EBV in patients with BL, which was 64.2% (95% CI: 52.0 to 75.6; *p* < 0.01) from 1969 to 1982, then 60.9% (95% CI: 50.3 to 71.1; *p* < 0.01) from 1983 to 1995, then 60.7% from 1996 to 2008, and finally 54.0% (95% CI: 42.2 to 65.5; *p* < 0.01) that had a lower prevalence than the pooled prevalence within the period from 2009 to 2021 (Table 3 and Appendix A). Furthermore, subgroup analyses based on the methods of EBV detection revealed a significantly increased prevalence when compared to the pooled prevalence in the nucleic acid hybridization at 81.7% (95% CI: 67.8 to 92.5; *p* < 0.01), 74.7% (95% CI: 60.0 to 87.1; *p* < 0.01) in the PCR method, and 60.0% (95% CI: 45.8 to 73.5; *p* < 0.01) in the immunofluorescence method. On the other hand, the prevalence in immunoassay, in situ hybridization (ISH), combined ISH with PCR, and Southern blot revealed a significantly lower prevalence: 54.7% (95% CI: 34.2 to 74.5; *p* < 0.01), 54.3% (95% CI: 46.3 to 62.1; *p* < 0.01), 53.2% (95% CI: 52.9 to 63.3; *p* = 0.01), and 47.1% (95% CI: 31.7 to 62.8; *p* < 0.01), respectively (Table 3 and Appendix A). The subgroup analysis based on different geographical locations revealed a significantly increased prevalence when compared to the pooled prevalence only in Sub-Saharan Africa and Northern Africa, at 76.5% (95% CI: 67.0 to 84.9; *p* < 0.01) and 69.3% (95% CI: 58.1 to 79.4; *p* < 0.01), respectively (Figure 3). Southern America, Southern Asia, Northern America, Europe, Eastern Asia, and South-eastern Asia showed a decrease in prevalence when compared to the pooled prevalence at 58.4% (95% CI: 50.0 to 66.6; *p* < 0.01), 54.7% (95% CI: 30.5 to 77.9; *p* = 0.12), 54.3% (95% CI: 34.5 to 73.5; *p* < 0.01), 49.5% (95% CI: 36.9 to 62.5; *p* < 0.01), 29.5% (95% CI: 19.9 to 40.1; *p* < 0.01), and 29.1% (95% CI: 11.0 to 51.2; *p* = 0.15), respectively (Table 3 and Appendix A). The subgroup analysis based on the socio-demographic index (SDI) revealed a significantly increased prevalence when compared to the pooled prevalence in both the middle and low SDI, at 60.1% (95% CI: 52.4 to 67.5; *p* < 0.01) and 82.7% (95% CI: 74.4 to 89.8; *p* = 0), respectively. On the other hand, countries with high SDI, high–middle SDI, and low–middle SDI showed a significant decrease in prevalence, at 43.0% (95% CI: 33.3 to 52.9; *p* = 0), 54.5% (95% CI: 40.0 to 68.6; *p* = 0), and 49.9% (95% CI: 31.4 to 68.5; *p* = 0), respectively (Table 3 and Appendix A).

### 3.5. Quality Assessment

In Appendix A, the quality assessment of the included studies was presented in detail. Generally, of the included studies, 65.2%, 29.6%, and 5.2% were high-, moderate-, and low-quality studies, respectively. The funnel plot and Egger’s test results revealed evidence of a publication bias for the prevalence of EBV in BL (*p* = 0.0034) (Figure 4).

### 3.6. Heterogeneity and Sensitivity Analysis

In sensitivity analyses, the highest EBV prevalence in patients with BL was observed when considering only case-control studies (67.6%; 95% CI: 58.0 to 76.5) [56,58,59,61,62,65,66,67,70,71,74,75,76,78,85,86,90,92,93,94,95,96,97,98,100,103,105,108,114,115,117,122,130,152,153,171,178,179,182], followed by considering only studies where the age was less than 18 years old (64.9%; 95% CI: 55.4 to 74.0) [56,58,59,61,62,63,69,70,72,73,74,75,77,78,81,89,90,94,95,98,99,102,103,104,105,107,108,109,110,113,115,116,117,120,122,126,135,137,138,163,165,166,169,174], excluding small studies with less than 100 subjects (64.0%; 95% CI: 40.3 to 84.9) [56,58,59,67,75,104,121,171], and excluding outlier studies (61.0%; 95% CI: 55.8 to 66.1) [56,57,58,59,60,61,62,63,64,65,66,67,68,69,70,71,72,73,74,75,76,77,78,79,80,81,82,83,84,85,86,87,88,89,90,91,92,93,94,95,96,97,98,99,100,101,102,103,104,105,106,107,108,109,110,111,112,113,114,115,116,117,118,119,120,121,122,123,124,125,126,127,128,129,130,131,132,133,134,135,136,137,138,139,140,141,142,143,144,145,146,147,148,149,150,151,152,153,154,155,156,157,158,159,160,161,162,164,166,167,168,169,170,171,172,173,174,175,176,178,179,180,181,182,183,184,185,186,187,188,189,190]. In contrast, the lowest EBV prevalence in patients with BL was found when considering only cohort studies (48.4%; 95% CI: 35.9 to 61.1) [64,89,99,110,120,125,126,138,142,149,154,161,167,169,176,177,181], followed by considering only cross-sectional studies (54.4%; 95% CI: 50.1 to 64.6) [57,60,63,68,69,72,73,77,79,80,81,82,83,84,87,88,91,101,102,104,106,107,109,111,112,113,116,118,119,121,123,124,127,128,129,131,132,133,134,135,136,137,139,140,141,143,144,145,146,147,148,150,151,155,156,157,158,159,160,162,163,164,165,166,168,170,172,173,174,180,183,184,185,186,187,188,189,190], and excluding low- and moderate-quality studies (58.7%; 95% CI: 51.8 to 65.3) [56,57,58,59,61,62,63,68,69,72,73,75,77,80,81,82,83,84,88,89,91,92,94,95,96,97,98,102,103,104,106,107,109,112,113,116,117,118,119,120,121,122,123,124,128,129,131,132,134,135,136,138,139,140,141,146,147,148,150,151,154,155,156,157,158,159,160,162,163,164,165,166,168,170,172,173,174,175,178,180,183,184,185,186,187,188,189,190] (Table 4 and Appendix A).

As depicted in the Galbraith plot (Figure 5), three outlier studies in estimating the prevalence of EBV in patients with BL were determined. The results showed significant heterogeneity at 97%, *p* < 0.001.

## 4. Discussion

EBV was found to be associated with human cancer when it was discovered in BL. This was a result of BL cell isolation. EBV has been extensively characterized due to purported links to a variety of human diseases, including BL, HL, post-transplant and AIDS-related lymphomas, and nasopharyngeal carcinoma [7,191,192]. Our results revealed a high prevalence of EBV in patients with BL, at 59.4% in all BL patients worldwide. However, as shown in our study, the prevalence of EBV in patients with BL varies by region; we found the highest prevalence in Sub-Saharan Africa (76.5%) and Northern Africa (69.3%), while the prevalence in Southern America (58.4%), Southern Asia (54.7%), Northern America (54.3%), Europe (49.7%), Eastern Asia (29.5%), and South-eastern Asia (29.1%) were lower than the pooled prevalence. We can explain the variations in EBV prevalence among patients with BL worldwide, as more than 95% of people in the world acquire the Epstein–Barr virus, a herpes virus belonging to the gamma subfamily, within the first ten years of life. Primary exposure to infections occurs in childhood in Africa and other developing countries, probably as a result of different cultural norms compared to developed countries [115,193].

The Epstein–Barr virus infection persists asymptomatically for the entirety of the host’s life, maintaining the immune system and this deceptive virus constantly in balance. In our study, the incidence of BL was higher in children (≤18) at 64.9% compared to adults; this corresponds to many studies that report that BL is more common in children [194,195]. Our results revealed that the incidence of BL among males is much higher than in females (34.7%), which is commensurate with several studies that report that BL is more prevalent in males compared to females [104,120,123,195]. This result is in agreement with Yakimchuk et al., which reported that estrogen has an anti-proliferative effect on BL cells through estrogen receptor β (ERβ) signalling [196]. Our study revealed a significant publication bias for EBV prevalence in patients with BL, and that is in agreement with some studies exploring the prevalence of EBV in different diseases, such as multiple sclerosis (*p* < 0.05) [197] and breast cancer (*p* = 0.006) [198], while that is in disagreement with some studies such as for gastric carcinoma (*p =* 0.912) [199], Hodgkin’s lymphoma (*p* = 0.162) [200], and EBV-associated epithelial tumors (*p* = 0.23617) [201].

Interestingly, our study shows a significant decline in EBV prevalence over four time periods (13 years), with the prevalence decreasing from 64.2% in the period from 1969 to 1982, to 54% in the period from 2009 to 2021. This decrease in incidence could be attributed to the development and widespread use of EBV vaccines, as well as improved sanitation, living habits, and personal hygiene [202,203]. There are many methods used to detect EBV, but these methods are different depending on whether they are faster, are more sensitive, or provide more informative than previous assays [204]. Our study revealed that the most used method in EBV detection was the microscopic examination (in situ hybridization (ISH) in 59 studies and immunofluorescence in 18 studies) method followed by molecular methods (PCR in 17 studies, nucleic acid hybridization in nine studies, ISH+PCR in nine studies, and Southern blot in seven studies), and, finally, immunoassay methods in seven studies. This result confirms that ISH is the methodology of choice for the detection of EBV in tissue sections [205,206,207]. Our results revealed a higher prevalence of EBV in patients with BL in both low and middle SDI countries, at 82.7% and 60.1%, respectively. A study showed that the highest incidence and mortality burden occurred in EBV-attributed BL in low and low–middle SDI areas [208]. The reasons for the increases in the burden of malignancies related to EBV infection appear to be growing populations, an increase in life expectancy, and changing age structure [209].

## 5. Conclusions

In conclusion, based on the comprehensive systematic and meta-analysis of the available data on the prevalence of EBV in patients with BL until January 2021, the prevalence was 59.4% in all patients with BL. Due to factors such as cultural habits, personality hygiene, limited use of developed EBV vaccines, and malaria endemic areas, Sub-Saharan Africa (76.5%) and Northern Africa (69.3%) revealed the highest prevalence (hot spots) in comparison to the rest of the world. Countries with middle and low SDI have a higher prevalence of EBV in patients with BL. Despite the fact that the EBV prevalence in patients with BL has decreased significantly from 64.2% in 1969 to 1982 to 54% from 2009 to 2021, as well as there being a higher incidence in younger (≤18) patients than adults, EBV detection should be used as a routine test in hot spots as well as in all young people because it will help in predicting whether populations are at a high risk of increasing the number of BL cases corresponding to EBV infection.

## Figures and Tables

**Figure 1 diagnostics-13-02068-f001:**
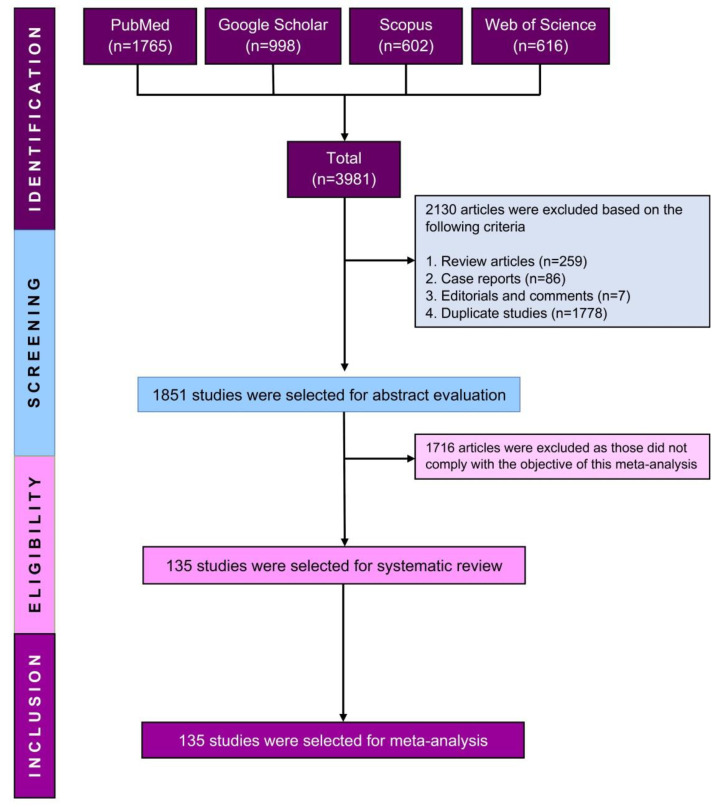
PRISMA flow diagram of study selection.

**Figure 2 diagnostics-13-02068-f002:**
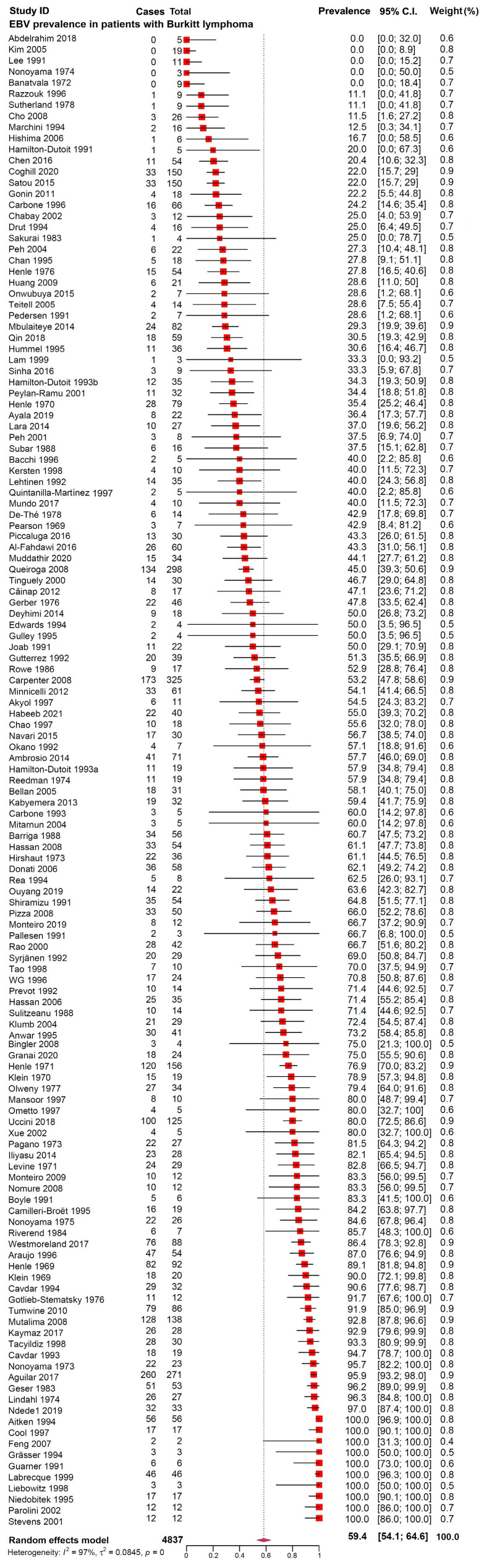
Forest plots presenting the prevalence of Epstein–Barr virus in patients with Burkitt lymphoma [55,56,57,58,59,60,61,62,63,64,65,66,67,68,69,70,71,72,73,74,75,76,77,78,79,80,81,82,83,84,85,86,87,88,89,90,91,92,93,94,95,96,97,98,99,100,101,102,103,104,105,106,107,108,109,110,111,112,113,114,115,116,117,118,119,120,121,122,123,124,125,126,127,128,129,130,131,132,133,134,135,136,137,138,139,140,141,142,143,144,145,146,147,148,149,150,151,152,153,154,155,156,157,158,159,160,161,162,163,164,165,166,167,168,169,170,171,172,173,174,175,176,177,178,179,180,181,182,183,184,185,186,187,188,189].

**Figure 3 diagnostics-13-02068-f003:**
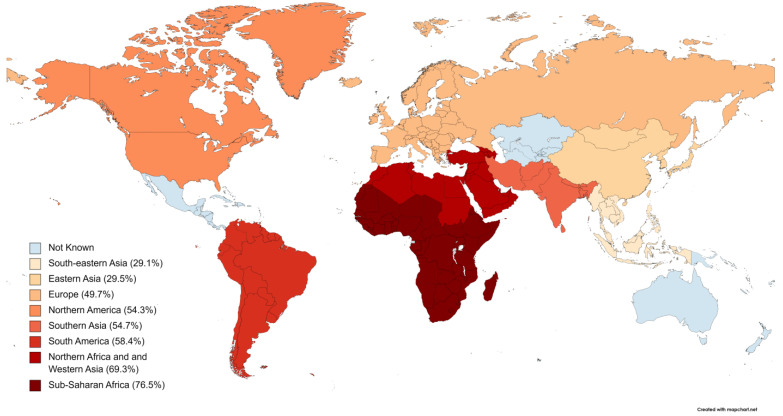
Global prevalence of Epstein–Barr virus in patients with Burkitt lymphoma.

**Figure 4 diagnostics-13-02068-f004:**
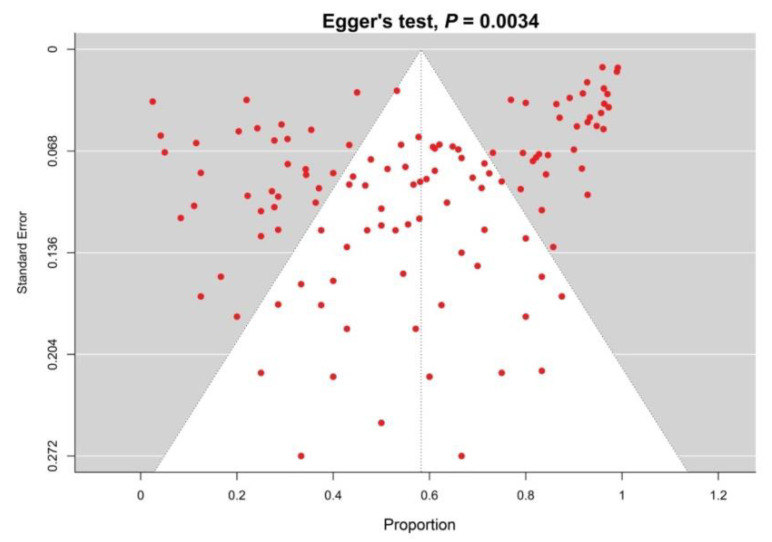
Funnel plots estimating the prevalence of EBV in patients with BL revealed significant publication bias [55,56,57,58,59,60,61,62,63,64,65,66,67,68,69,70,71,72,73,74,75,76,77,78,79,80,81,82,83,84,85,86,87,88,89,90,91,92,93,94,95,96,97,98,99,100,101,102,103,104,105,106,107,108,109,110,111,112,113,114,115,116,117,118,119,120,121,122,123,124,125,126,127,128,129,130,131,132,133,134,135,136,137,138,139,140,141,142,143,144,145,146,147,148,149,150,151,152,153,154,155,156,157,158,159,160,161,162,163,164,165,166,167,168,169,170,171,172,173,174,175,176,177,178,179,180,181,182,183,184,185,186,187,188,189].

**Figure 5 diagnostics-13-02068-f005:**
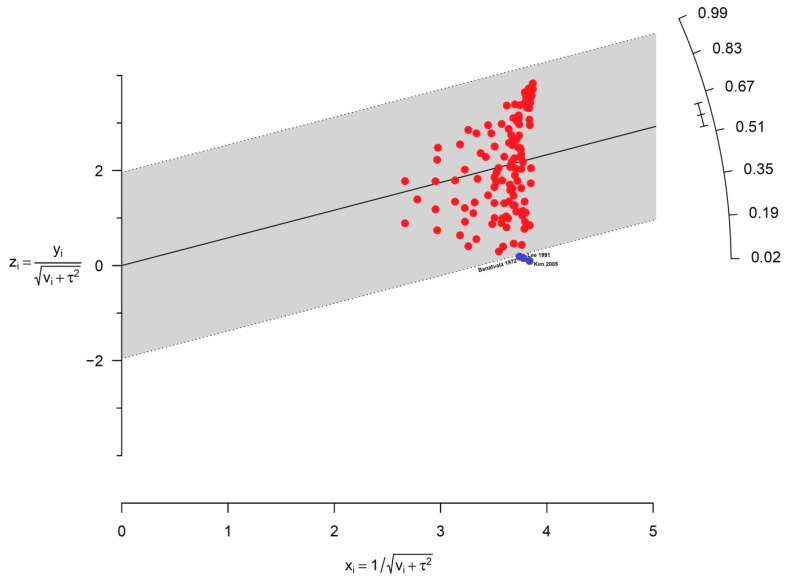
Galbraith plots show two outlier studies in estimating the prevalence of EBV in patients with BL [55,56,57,58,59,60,61,62,63,64,65,66,67,68,69,70,71,72,73,74,75,76,77,78,79,80,81,82,83,84,85,86,87,88,89,90,91,92,93,94,95,96,97,98,99,100,101,102,103,104,105,106,107,108,109,110,111,112,113,114,115,116,117,118,119,120,121,122,123,124,125,126,127,128,129,130,131,132,133,134,135,136,137,138,139,140,141,142,143,144,145,146,147,148,149,150,151,152,153,154,155,156,157,158,159,160,161,162,163,164,165,166,167,168,169,170,171,172,173,174,175,176,177,178,179,180,181,182,183,184,185,186,187,188,189].

**Table 1 diagnostics-13-02068-t001:** Epidemiology features of EBV-associated neoplasia.

Gene Expressed	Product	EBV Latency Programs				
		Latency O	Latency I	Latency IIa	Latency IIb	Latency III
EBNA1	Protein	–	+	+	+	+
EBNA2	Protein	–	+	+	+	+
EBNA3	Protein	–	–	+	+	+
EBNA-LP	Protein	–	–	+	+	+
LMP1	Protein	–	–	+	+	+
LMP2	Protein	–	–	+	+	+
BARTs	Protein	–	–	+	+	+
EBERs	ncRNAs	+	+	+	+	+
**Associated Malignancies**		Memory B cells	- BL - EBVaGC	- NPC - T/NK LPD - EBV + DLBCL, NOS - cHL - NLPHL		- AIDS-associated B-cell lymphoma - DLBCL - PTLD

EBNA: EB viral nuclear antigen; EBNA-LP: EB viral nuclear antigen leader protein; LMP: latent membrane protein; BARTs: BamHI A rightward transcripts; EBERs: Epstein–Barr virus-encoded small RNAs; ncRNAs: non-coding RNAs; BL: Burkitt Lymphoma; EBVaGC: Epstein–Barr virus-associated gastric cancer; NPC: nasopharyngeal cancer; LPD: lymphoproliferative disorder; DLBCL, NOS: diffuse large B-cell lymphoma, not otherwise specified; cHL: classic Hodgkin’s Lymphoma; NLPHL: nodular lymphocyte-predominant Hodgkin’s lymphoma;PTLD: post-transplant lymphoproliferative disorder. “+” indicates the protein is expressed, while “−” indicates that the protein is not expressed.

**Table 2 diagnostics-13-02068-t002:** Major characteristics of the included studies.

No	Study ID	Type of Study	Country	Age (Mean ± SD/Range) (Years)	Type of Participants		Number of BL Patients (% Female)	EBV Positivity in BL # (%)	**Sample Type**	**Method of Detection of EBV**
					Age Group	# (%)				
1	Abdelrahim, 2018 [175]	Cross-sectional	Malaysia	22.2 ± 14.5 (32)	≤18	3 (60)	5 (60)	0 (0)	FFPE	ISH
					45 < Age > 18	2 (40)				
2	Aguilar, 2017 [56]	Case-control	Malawi	7.8 ± 2.9 (6.23)	≤18	271 (100)	271 (41)	260 (95.9)	Sera	qSAT
3	Aitken, 1994 [57]	Cross-sectional	New Guinea	NR	NR	NR	56 (NR)	56 (100)	FFPE	PCR
4	Akyol, 1997 [91]	Cross-sectional	Turkey	17 ± 19.1 (59)	≤18	7 (63.6)	11 (18.2)	6 (54.5)	FFPE	ISH
					45 < Age > 18	1 (9.1)				
					≥45	1 (9.1)				
					NR	2 (18.5)				
5	Al-Fahdawi, 2016 [92]	Case-control	Iraq	21.8 ± 11.7 (34)	≤18	19 (31.7)	60 (31.7)	26 (43.3)	FFPE	PCR
					45 < Age > 18	14 (68.3)				
6	Ambrosio, 2014 [181]	Cohort	Kenya, Spain, and Italy	NR	NR	NR	71 (NR)	41 (57.7)	FFPE	ISH
7	Anwar, 1995 [93]	Case-control	Egypt	9.1 ± 4.9 (20)	≤18	38 (92.7)	41 (41.5)	30 (73.2)	FFPE and fresh tumor biopsies	ISH
					45 < Age > 18	3 (7.3)				
8	Araujo, 1996 [105]	Case-control	Brazil	5.9 ± 3 (13)	≤18	54 (100)	54 (34.6)	47 (87)	FFPE	ISH
9	Ayala, 2019 [184]	Cross-sectional	Kenya and Italy	16.6 ± 13.2 (42)	≤18	15 (68.2)	22 (50)	8 (36.4)	FFPE	ISH
					45 < Age > 18	6 (27.3)				
					≥45	1 (4.5)				
10	Bacchi, 1996 [106]	Cross-sectional	Brazil	36 ± 8.5 (21)	45 < Age > 18	5 (100)	5 (20)	2 (40)	FFPE	ISH
11	Banatvala, 1972 [177]	Cohort	East Africa	NR	NR	NR	9 (NR)	0 (0)	Sera	Immunofluorescence
12	Barriga, 1988 [185]	Cross-sectional	Ghana and USA	NR	NR	NR	56 (NR)	34 (60.7)	Fresh tumor biopsies	Southern blotting
13	Bellan, 2005 [182]	Case-control	Kenya, France, and Italy	Range, 66	NR	NR	31 (NR)	18 (58.1)	FFPE	ISH
14	Bingler, 2008 [126]	Cohort	USA	2.1 ± 2.5 (5.1)	≤18	4 (100)	4 (50)	3 (75)	FFPE	PCR
15	Boyle, 1991 [176]	Cohort	Australia	NR	NR	NR	7 (NR)	5/6 (83.3)	FFPE and fresh tumor biopsies	PCR
16	Căinap, 2012 [138]	Cohort	Romania	NR	≤18	17 (100)	17 (NR)	8 (47.1)	Sera	Serological IgG VCA antibody
17	Camilleri-Broët, 1995 [139]	Cross-sectional	France	NR	NR	NR	19 (NR)	16 (84.2)	FFPE	PCR
18	Carbone, 1993 [141]	Cross-sectional	Italy	NR	NR	NR	5 (NR)	3 (60)	FFPE	ISH
19	Carbone, 1996 [140]	Cross-sectional	Italy	NR	NR	NR	66 (NR)	16 (24.2)	Fresh tumor biopsies	ISH
20	Carpenter, 2008 [58]	Case-control	Uganda	7 ± 3 (13.5)	≤18	325 (100)	325 (39.1)	173 (53.2)	Sera	Chemiluminescent immunoassay
21	Cavdar, 1993 [94]	Case-control	Turkey	5.5 (12)	≤18	72 (100)	72 (31.9)	18/19 (94.7)	Fresh tumor biopsies	Southern blotting + PCR
22	Cavdar, 1994 [95]	Case-control	Turkey	Median 5 (4.5)	≤18	81 (100)	81 (30)	29/32 (90.6)	Fresh tumor biopsies	Immunofluorescence
23	Chabay, 2002 [107]	Cross-sectional	Argentina	Range, 13.75	≤18	12 (100)	12 (NR)	3 (25)	FFPE	ISH + PCR
24	Chan, 1995 [156]	Cross-sectional	China	40.8 ± 24.9 (81)	≤18	10 (55.6)	18 (44.4)	5 (27.8)	FFPE	ISH
					45 < Age > 18	8 (44.4)				
25	Chao, 1997 [157]	Cross-sectional	Taiwan	33 ± 24.3 (72)	≤18	6 (33.3)	18 (33.3)	10 (55.6)	FFPE	ISH
					45 < Age > 18	7 (38.9)				
					≥45	5 (27.8)				
26	Chen, 2016 [158]	Cross-sectional	Taiwan	Median 27 (82)	≤18	21 (38.9)	54 (33)	11 (20.4)	FFPE	ISH
					NR	33 (66.1)				
27	Cho, 2008 [159]	Cross-sectional	South Korea	36 (NR)	NR	NR	26 (38.5)	3 (11.5)	FFPE	ISH
28	Coghill, 2020 [59]	Case-control	Ghana	8.3 (17)	≤18	150 (100)	150 (36.7)	33 (22.0)	Sera	Microarray
29	Cool, 1997 [60]	Cross-sectional	Kenya	Range, 56	NR	NR	21 (NR)	17/17 (100)	FFPE	ISH
30	De-Thé, 1978 [61]	Case-control	Uganda	6.6 ± 2.6 (9)	≤18	14 (100)	14 (35.7)	6 (42.9)	Fresh tumor biopsies	Nucleic acid hybridization
31	Deyhimi, 2014 [123]	Cross-sectional	Iran	21 (79)	NR	NR	18 (27.8)	9 (50)	FFPE	ISH + PCR
32	Donati, 2006 [110]	Cohort	Brazil	6.2 (NR)	≤18	58 (100)	58 (34.5)	36 (62.1)	Fresh tumor biopsies	HIS-FISH
33	Drut, 1994 [109]	Cross-sectional	Argentina	NR	≤18	16(100)	16 (50)	4 (25.0)	FFPE	PCR
34	Edwards, 1994 [127]	Cross-sectional	USA	NR	NR	NR	4 (NR)	2 (50)	Fresh tumor biopsies	PCR
35	Feng, 2007 [160]	Cross-sectional	China	Median 18.5 (31)	≤18	1 (50)	2 (50)	2 (100)	FFPE	Nucleic acid hybridization
					45 < Age > 18	1 (50)				
36	Gerber, 1976 [62]	Case-control	Ghana	Range, 12	≤18	46 (100)	46 (NR)	22 (47.8)	Sera	Immunofluorescence
37	Geser, 1983 [63]	Cross-sectional	Uganda and Sudan	7.5(15)	≤18	74 (100)	74 (80)	51/53 (96.2)	Fresh tumor biopsies	Nucleic acid hybridization
38	Gonin, 2011 [142]	Cohort	France	NR	NR	NR	18 (NR)	4 (22.2)	FFPE	ISH
39	Gotlieb- Stematsky, 1976 [96]	Case-control	Israel	7.2 ± 4.9 (16)	≤18	15 (93.8)	16 (37.5)	11/12 (91.7)	Sera	Immunofluorescence
					45 < Age > 18	1 (6.2)				
40	Granai, 2020 [64]	Cohort	Uganda	NR	NR	NR	24 (NR)	18 (75)	FFPE	IHC
41	Grässer, 1994 [143]	Cross-sectional	UK	NR	NR	NR	3 (NR)	3 (100)	FFPE and fresh tumor biopsies	ISH
42	Guarner, 1991 [128]	Cross-sectional	USA	36.7 ± 3.9 (10)	45 < Age > 18	6 (100)	6 (NR)	6 (100)	FFPE	ISH
43	Gulley, 1995 [129]	Cross-sectional	USA	35 (46)	NR	NR	4 (50)	2 (50)	FFPE	ISH
44	Gutterrez, 1992 [111]	Cross-sectional	South America (Brazil, Chile, and Argentina)	7.4 ± 5.1 (27)	≤18	37 (94.9)	39 (18)	20 (51.3)	Fresh tumor biopsies	Southern blotting
					45 < Age > 18	2 (5.1)				
45	Habeeb, 2021 [97]	Case-control	Syria	11.5 (56)	4 to 12	37 (92.5)	40 (27.5)	22 (55)	FFPE	ISH
					48-60	3 (7.5)				
46	Hamilton-Dutoit, 1991 [112]	Cross-sectional	Argentina	45 ± 18.5 (59)	45 < Age > 18	4 (57.1)	7 (14.3)	1/5 (20)	FFPE	ISH
					≥45	3 (42.9)				
47	Hamilton-Dutoit, 1993a [144]	Cross-sectional	Denmark	37.2 ± 13.1 (64)	≤18	1 (5.3)	19 (0)	11 (58)	Fresh tumor biopsies	Southern blotting
					45 < Age > 18	5 (26.3)				
					≥45	13 (68.4)				
48	Hamilton-Dutoit, 1993b [145]	Cross-sectional	Denmark	NR	NR	NR	35 (NR)	12 (34.3)	FFPE	ISH
49	Hassan, 2006 [114]	Case-control	Brazil	Range, 14	NR	NR	35 (NR)	25 (71.4)	FFPE	ISH+PCR
50	Hassan, 2008 [113]	Cross-sectional	Brazil	Median 5 (12)	≤18	54 (100)	54 (33.3)	33 (61.1)	FFPE and fresh tumor biopsies	ISH + PCR
51	Henle, 1969 [66]	Case-control	Kenya	NR	NR	NR	92 (NR)	82 (89.1)	Sera	Immunofluorescence
52	Henle, 1970 [76]	Case-control	Kenya	NR	NR	NR	79 (NR)	28 (35.4)	Fresh tumor biopsies	Immunofluorescence
53	Henle, 1971 [67]	Case-control	Kenya	NR	NR	NR	156 (NR)	120 (76.9)	Sera	Immunofluorescence
54	Henle, 1976 [65]	Case-control	Uganda and Ghana	NR	NR	NR	54 (NR)	15 (27.8)	Sera	Immunofluorescence
55	Hirshaut, 1973 [178]	Case-control	Uganda and USA	NR	NR	NR	36 (NR)	22 (61.1)	Sera	Immunofluorescence
56	Hishima, 2006 [161]	Cohort	Japan	34.7 (16)	45 < Age > 18	6 (100)	6 (0)	1 (16.7)	Fresh tumor biopsies	ISH
57	Huang, 2009 [162]	Cross-sectional	China	27.8 ± 20.6 (69)	≤18	7 (33.3)	21 (19.0)	6 (28.6)	FFPE	ISH
					45 < Age > 18	10 (47.6)				
					≥45	4 (19.1)				
58	Hummel, 1995 [146]	Cross-sectional	Germany	NR	NR	NR	36 (NR)	11 (30.6)	FFPE	ISH
59	Iliyasu, 2014 [68]	Cross-sectional	Nigeria	NR	NR	NR	28 (NR)	23 (82.1)	FFPE	ISH
60	Joab, 1991 [179]	Case-control	France & China	NR	NR	NR	22 (NR)	11 (50)	Sera	Immunofluorescence
61	Kabyemera, 2013 [98]	Case-control	Tanzania	Range, 14	≤18	32 (100)	32 (56.2)	19 (59.4)	Blood	PCR
62	Kaymaz, 2017 [69]	Cross-sectional	Kenya	Median 8.2 (12)	≤18	28 (100)	28 (29.0)	26 (92.9)	Fresh tumor biopsies	PCR
63	Kersten, 1998 [147]	Cross-sectional	Netherlands	NR	NR	NR	10 (NR)	4 (40)	FFPE	ISH
64	Kim, 2005 [163]	Cross-sectional	South Korea	Range, 17.2	≤18	19 (100)	19 (NR)	0 (0)	FFPE	ISH
65	Klein, 1969 [71]	Case-control	Kenya	NR	NR	NR	20 (NR)	18 (90)	Sera	Immunofluorescence
66	Klein, 1970 [70]	Case-control	Kenya	7.2 ± 3.0 (12)	≤18	19 (100)	19 (42.1)	15 (78.9)	Sera	Immunofluorescence
67	Klumb, 2004 [115]	Case-control	Brazil	Range, 8	≤18	37 (100)	37 (32.4)	21/29 (72.4)	FFPE and fresh tumor biopsies	PCR
68	Labrecque, 1999 [72]	Cross-sectional	Malawi	7.1 (10)	≤18	46 (100)	46 (39.1)	46 (100)	FFPE and fresh tumor biopsies	ISH
69	Lam, 1999 [164]	Cross-sectional	China	47.7 ± 31.8 (61)	≤18	2 (66.7)	3 (100)	1 (33.3)	FFPE	ISH
					≥45	1 (33.3)				
70	Lara, 2014 [116]	Cross-sectional	Argentina	Range, 15	≤18	27 (100)	27 (37.0)	10 (37.0)	FFPE	ISH
71	Lee, 1991 [165]	Cross-sectional	Taiwan	Range, 14	≤18	11 (100)	11 (NR)	0 (0)	FFPE	Southern blotting
72	Lehtinen, 1992 [183]	Cross-sectional	Finland and Tanzania	28.3 (65)	NR	NR	35 (42.9)	14 (40)	FFPE	ISH
73	Levine, 1971 [130]	Case-control	USA	13.7 ± 9.2 (40)	≤18	23 (79.3)	29 (41.4)	24 (82.8)	Sera	Immunofluorescence
					45 < Age > 18	6 (20.7)				
74	Liebowitz, 1998 [131]	Cross-sectional	USA	NR	NR	NR	3 (NR)	3 (100)	Fresh tumor biopsies	PCR
75	Lindahl, 1974 [73]	Cross-sectional	Africa	7.6 ± 2.9 (10)	≤18	27 (100)	27 (44.4)	26 (96.3)	FFPE and fresh tumor biopsies	Nucleic acid hybridization
76	Mansoor, 1997 [124]	Cross-sectional	Pakistan	10.7 ± 5.7 (18)	≤18	9 (90)	10 (30)	8 (80)	FFPE	ISH
					45 < Age > 18	1 (10)				
77	Marchini, 1994 [148]	Cross-sectional	Sweden	NR	NR	NR	16 (NR)	2 (12.5)	Sera	ELISA
78	Mbulaiteye, 2014 [132]	Cross-sectional	USA	NR	0–19	24 (26)	91 (13)	24/82 (29.3)	FFPE	ISH
					20–34	14 (15)				
					35–59	26 (29)				
					≥60	17 (19)				
					NR	10 (11)				
79	Minnicelli, 2012 [117]	Case-control	Brazil	Median 5 (12)	≤18	62 (100)	62 (30.6)	33/61 (54.1)	FFPE	ISH
80	Mitarnun, 2004 [172]	Cross-sectional	Thailand	35.6 (31)	45 < Age > 18	5 (100)	5 (40)	3 (60)	FFPE	ISH + PCR
81	Monteiro, 2009 [118]	Cross-sectional	Brazil	Range, 95	≤15	7/10 (70)	12 (33.3)	10 (83.3)	FFPE	ISH
					>15	3/10 (30)				
82	Monteiro, 2019 [119]	Cross-sectional	Brazil	23,8 (95)	NR	NR	12 (33.3)	8/12 (66.7)	FFPE	ISH
83	Muddathir, 2020 [74]	Case-control	Sudan	Range, 11	≤18	34 (100)	34 (38.2)	15 (44.1)	FFPE	IHC
84	Mundo, 2017 [149]	Cohort	Italy	14.2 (38)	NR	NR	10 (50)	4 (40)	FFPE	ISH
85	Mutalima, 2008 [75]	Case-control	Malawi	7.1 ± 2.6 (15)	≤18	148 (100)	148 (40)	128/138 (92.8)	Sera	Immunofluorescence
86	Navari, 2015 [189]	Cross-sectional	Italian and African	35 ± 22.8 (74.5)	≤18	8 (26.7)	30 (7/20 [35%])	17 (56.7)	FFPE	DASL
					45 < Age > 18	9 (30)				
					≥45	10 (33.3)				
					NR	3 (10)				
87	Ndede1, 2019 [77]	Cross-sectional	Kenya	NR	≤18	33 (100)	33 (21.2)	32 (97)	Sera + fresh tumor biopsies	ELISA + IHC
88	Niedobitek, 1995 [78]	Case-control	Uganda and Malawi	8.2 ± 3.9 (14)	≤18	17 (100)	17 (53)	17 (100)	FFPE	ISH
89	Nomure, 2008 [166]	Cross-sectional	Japan	6.2 ± 2.7 (10)	≤18	12 (100)	12 (25)	10 (83.3)	FFPE	ISH
90	Nonoyama, 1973 [79]	Cross-sectional	Kenya	NR	NR	NR	23 (NR)	22 (95.7)	Fresh tumor biopsies	Nucleic acid hybridization
91	Nonoyama, 1974 [133]	Cross-sectional	USA	NR	NR	NR	3 (NR)	0 (0)	Fresh tumor biopsies	Nucleic acid hybridization
92	Nonoyama, 1975 [80]	Cross-sectional	Kenya	NR	NR	NR	26 (NR)	22 (84.6)	Fresh tumor biopsies	Nucleic acid hybridization
93	Okano, 1992 [167]	Cohort	Japan	14.7 ± 12.7 (35)	≤18	6 (85.7)	7 (42.9)	4 (57.1)	Fresh tumor biopsies	Southern blotting
					45 < Age > 18	1 (14.3)				
94	Olweny, 1977 [81]	Cross-sectional	Uganda	7.2 (13)	≤18	34 (100)	34 (32.3)	27 (79.4)	Fresh tumor biopsies	Nucleic acid hybridization
95	Ometto, 1997 [150]	Cross-sectional	Italy	NR	NR	NR	5 (NR)	4 (80.0)	FFPE	PCR
96	Onwubuya, 2015 [82]	Cross-sectional	Nigeria	16.9 (50)	0–20	6 (85.7)	7 (28.6)	2 (28.6)	FFPE	ISH
					41–60	1 (14.3)				
97	Ouyang, 2019 [168]	Cross-sectional	China	NR	NR	NR	22 (NR)	14 (63.6)	FFPE	ISH
98	Pagano, 1973 [134]	Cross-sectional	USA	NR	NR	NR	27 (NR)	22 (81.5)	Fresh tumor biopsies	Nucleic acid hybridization
99	Pallesen, 1991 [151]	Cross-sectional	Denmark	39.3 ± 6.4 (12)	45 < Age > 18	8 (61.5)	3 (0)	2 (66.7)	Fresh tumor biopsies	ISH
100	Parolini, 2002 [152]	Case-control	Italy	NR	NR	NR	12 (0)	12 (100)	FFPE	ISH
101	Pearson, 1969 [153]	Case-control	Sweden	NR	NR	NR	7 (NR)	3 (37.5)	Sera	Immunofluorescence
102	Pedersen, 1991 [154]	Cohort	Denmark	NR	NR	NR	12 (NR)	2/7 (28.6)	FFPE	ISH
103	Peh, 2001 [173]	Cross-sectional	Malaysia	NR	NR	NR	8 (0)	3 (37.5)	FFPE	ISH
104	Peh, 2004 [174]	Cross-sectional	Malaysia	Range, 15	≤18	22 (100)	22 (22.7)	6 (27.3)	FFPE	ISH
105	Peylan-Ramu, 2001 [99]	Cohort	Israel	Median, 5	≤18	32 (100)	32 (25)	11 (34.4)	FFPE	ISH
106	Piccaluga, 2016 [190]	Cross-sectional	Italy and Africa	NR	NR	NR	30 (NR)	13 (43.3)	FFPE	DASL
107	Pizza, 2008 [120]	Cohort	Brazil	6±2.7 (13)	≤18	53 (100)	53 (24.5)	33/50 (66.0)	FFPE	ISH
108	Prevot, 1992 [83]	Cross-sectional	Cameroon, Gabon	NR	NR	NR	14 (NR)	10 (83.3)	FFPE	ISH
109	Qin, 2018 [169]	Cohort	China	NR	≤18	105 (100)	105 (15.2)	18/59 (30.5)	Fresh tumor biopsies	ISH
110	Queiroga, 2008 [121]	Cross-sectional	Brazil	23.1 (93)	≤16	149 (47.9)	311 (28.9)	134/298 (45)	FFPE	ISH
					>16	143 (46)				
					NR	19 (6.1)				
111	Quintanilla-Martínez, 1997 [187]	Cross-sectional	Mexico and European	NR	NR	NR	5 (NR)	2 (40)	FFPE	ISH + PCR
112	Rao, 2000 [188]	Cross-sectional	Southern India and Argentina	7.4 ± 5.1 (23)	≤18	39 (92.9)	42 (33.3)	28 (66.7)	FFPE	ISH + PCR
					45 < Age > 18	2 (4.7)				
					NR	1 (2.4)				
113	Razzouk, 1996 [135]	Cross-sectional	USA	Range, 13	≤18	9 (100)	9 (33.3)	1 (11.1)	Fresh tumor biopsies	Immunofluorescence
114	Rea, 1994 [155]	Cross-sectional	France	NR	NR	NR	9 (NR)	5/8 (62.5)	FFPE	ISH
115	Reedman, 1974 [84]	Cross-sectional	Kenya	8.7 ± 5.3 (19)	≤18	16 (84.2)	19 (NR)	11/19 (57.9)	Fresh tumor biopsies	CF
					45 < Age > 18	2 (10.5)				
					NR	1 (5.3)				
116	Riverend, 1984 [122]	Case-control	Cuba	7.6 ± 3.3 (9)	≤18	7 (100)	7 (42.9)	6 (85.7)	FFPE	CF
117	Rowe, 1986 [180]	Cross-sectional	France, Algeria, La Rcunion, and England	12.1±13.4 (51)	≤18	14 (82.3)	17 (29.4)	9 (53)	Fresh tumor biopsies	Immunofluorescence
					45 < Age > 18	2 (11.8)				
					≥45	1 (5.9)				
118	Sakurai, 1983 [170]	Cross-sectional	Japan	15.8 ± 16.7 (41)	≤18	4 (80)	5 (40)	1/4 (25)	Fresh tumor biopsies	ELISA
					≥45	1 (20)				
119	Satou, 2015 [171]	Case-control	Japan	Range, 85	NR	NR	150 (20.7)	33 (22)	FFPE	ISH
120	Shiramizu, 1991 [186]	Cross-sectional	Ghana & USA	NR	NR	NR	54 (NR)	35 (64)	Fresh tumor biopsies	Southern blotting
121	Sinha, 2016 [125]	Cohort	India	NR	≤18	7 (77.7)	9 (NR)	3 (33.3)	Plasma	PCR
					NR	3 (33.3)				
122	Stevens, 2001 [85]	Case-control	Malawi	NR	NR	NR	12 (NR)	12 (100)	Blood	PCR
123	Subar, 1988 [136]	Cross-sectional	USA	NR	NR	NR	16 (NR)	6 (37.5)	Fresh tumor biopsies	Southern blotting
124	Sulitzeanu, 1988 [100]	Case-control	Israel	NR	NR	NR	14 (NR)	10 (71.4)	sera	LMI
125	Sutherland, 1978 [86]	Case-control	Uganda	NR	NR	NR	9 (NR)	1 (11.1)	Fresh tumor biopsies	Immunofluorescence
126	Syrjänen, 1992 [101]	Cross-sectional	Tanzania	Range, 15	NR	NR	29 (14/27 [51.9%])	20 (69)	FFPE	PCR
127	Tacyildiz, 1998 [102]	Cross-sectional	Turkey	5.9 (NR)	≤18	30 (100)	30 (NR)	28 (93.3)	FFPE	PCR
128	Tao, 1998 [87]	Cross-sectional	Ghana	NR	NR	NR	10 (NR)	7 (70)	Fresh tumor biopsies	PCR
129	Teitell, 2005 [137]	Cross-sectional	USA	8.9 ± 4.6 (14)	≤18	14 (100)	14 (14.3)	4 (28.6)	FFPE	ISH + PCR
130	Tinguely, 2000 [103]	Case-control	Turkey	4.8 (9.5)	≤18	30 (100)	30 (NR)	14 (46.7)	FFPE	ISH+PCR
131	Tumwine, 2010 [88]	Cross-sectional	Uganda	NR	NR	NR	86 (NR)	79(91.9)	FFPE	ISH
132	Uccini, 2018 [104]	Cross-sectional	Iraq	5.9 ± 3.1	≤18	125 (100)	125 (21.1)	100 (80)	FFPE	ISH
133	Westmoreland, 2017 [89]	Cohort	Malawi	9.3±3.8	≤18	88 (100)	88 (34.1)	76 (86.4)	Fresh tumor biopsies and sera	ISH
134	WG, 1996 [108]	Case-control	Brazil	Median, 6	≤18	13/24 (54.1)	24 (8/15 [53.3%])	17 (70.8)	FFPE	ISH
135	Xue, 2002 [90]	Case-control	Malawi	7 ± 2.4 (6)	≤18	7 (100)	7 (57.1)	4/5 (80)	Fresh tumor biopsies	ISH

NR: not reported; #: number of cases; FFPE: formalin-fixed paraffin-embedded; ISH: in situ hybridization; qSAT: quantitative suspension array technology; PCR: polymerase chain reaction; HIS-FISH: histology fluorescence in situ hybridization; IHC: immunohistochemistry; ELISA: enzyme-linked immunosorbent assay; DASL: cDNA-mediated annealing, selection, extension, and ligation; CF: complement fixation; and LMI: leukocyte migration inhibition.

**Table 3 diagnostics-13-02068-t003:** Subgroup analysis of prevalence of EBV in patients with BL.

Subgroups	Prevalence of EBV [95% CI]	Studies Number	Positive for EBV	Heterogeneity	
				I^2^, %	*p* Value
Time Interval Trend					
From 1969 to 1982	64.2 [52.0–75.6]	21	497	95.0	<0.01
From 1983 to 1995	60.9 [50.3–71.1]	37	473	95.0	<0.01
From 1996 to 2008	60.7 [51.7–69.3]	43	939	97.0	<0.01
From 2009 to 2021	54.0 [42.2–65.5]	34	1005	98.0	<0.01
Methods of EBV detection					
Nucleic acid hybridization	81.7 [67.8–92.5]	9	178	86.0	<0.01
Polymerase chain reaction (PCR)	74.7 [60.0–87.1]	17	255	91.0	<0.01
Immunofluorescence	60.0 [45.8–73.5]	18	539	96.0	<0.01
Immunoassay	54.7 [34.2–74.5]	7	453	90.0	<0.01
In situ hybridization (ISH)	54.3 [46.3–62.1]	59	1058	97.0	<0.01
ISH+PCR	53.2 [52.9–63.3]	9	121	60.0	0.01
Southern blot	47.1 [31.7–62.8]	7	110	92.0	<0.01
Geographical location					
Sub-Saharan Africa	76.5 [67.0–84.9]	35	1500	77.0	<0.01
Northern Africa	69.3 [58.1–79.4]	14	344	89.0	<0.01
Southern America	58.4 [50.0–66.6]	18	443	84.0	<0.01
Southern Asia	54.7 [30.5–77.9]	3	20	66.0	0.05
Northern America	54.3 [34.5–73.5]	12	97	84.0	<0.01
Europe	49.7 [36.9–62.5]	18	122	91.0	<0.01
Eastern Asia	29.5 [19.9–40.1]	16	119	86.0	<0.01
South-eastern Asia	29.1 [11.0–51.2]	4	12	62.0	0.05
Socio-demographic Index					
High SDI	43.0 [33.3–52.9]	35	250	83.0	0
High–middle SDI	54.5 [40.0–68.6]	21	201	87.0	0
Middle SDI	60.1 [52.4–67.5]	25	641	82.0	<0.01
Low–middle SDI	49.9 [31.4–68.5]	8	115	87.0	0
Low SDI	82.7 [74.4–89.8]	28	1343	94.0	0

BL: Burkitt lymphoma; EBV: Epstein-Barr virus; CI: Confidence interval; SCI: socio-demographic index.

**Table 4 diagnostics-13-02068-t004:** Sensitivity analyses.

Strategies of Sensitivity Analyses	Prevalence [95% CIs] (%)	Difference of Pooled Prevalence Compared to the Main Result	Number of Studies Analyzed	Total Number of Subjects	Heterogeneity	
					I^2^, %	*p* Value
Excluding small studies (<100)	64.0 [40.3–84.9]	4.6% higher	8	1613	99%	<0.001
Excluding low- and moderate-quality studies	58.7 [51.8–65.3]	0.7 lower	88	3383	93%	<0.01
Considering only cross-sectional studies	54.4 [50.1–64.6]	5% lower	79	2114	91%	<0.01
Considering only case-control studies	67.6 [58.0–76.5]	8.2% higher	39	2218	97%	<0.01
Considering only cohort studies	48.4 [35.9–61.1]	11% lower	17	475	85%	<0.01
Considering only studies where the age was less than 18 years old	64.9 [55.4–74.0]	5.5% higher	44	2187	95%	<0.01
Excluding outlier studies	61.0 [55.8–66.1]	1.6% higher	132	4798	92%	<0.01

CIs: confidence intervals.

## Data Availability

The data are contained within the article or Appendix A.

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
