# Peer review of "Worldwide Prevalence of Epstein–Barr Virus in Patients with Burkitt Lymphoma: A Systematic Review and Meta-Analysis"

_diagnostics, 2023, doi:10.3390/diagnostics13122068_

Round 1

Reviewer 1 Report

The manuscript by Mutaz Jamal Al-Hreisat et al. presents a systematic review aimed at assessing the prevalence of Epstein-Barr virus in patients with Burkitt's lymphoma. 135 studies published between 1969 and 202, which examined the prevalence of EBV in patients with BL were included in the study for meta-analysis. The authors present data from 4837 patients with BL lymphoma. The rationale for this type of research is not entirely clear, nor are the results. Is 57.5% the median value for the three clinical BL variants, or what does it reflect?

Three clinical variants of BL have been described based on the epidemiology of the cancer: endemic (eBL), sporadic (sBL) and immunodeficiency-associated (ID BL).

These variants are similar in morphology, immunophenotype and genetics. While sBL occurs outside of Africa and is rarely associated with EBV infection, eBL arises mainly in Africa and is associated with malaria endemicity and EBV infection.  The epidemiology for BL is different among various geographic areas. Differences in BL reflect distinct pathogenesis of the disease, and this well-known aspect could be valid. In addition, the other epidemiological factors and/or possible genetic predispositions still have an unclear role in the genesis of BL. To the present day, the exact cellular and molecular events that link EBV to increase the risk for BL remain to be elucidated.

EBV is detected in ~95% of EPD cases. EBV can also be found in about 30% of cases of sporadic Burkitt's lymphoma and in 25-40% of cases of immunodeficiency-associated Burkitt's lymphoma. To get the median, we sum up 95+30+40=165/3=55%.

Author Response

Reviewer 1

The manuscript by Mutaz Jamal Al-Hreisat et al. presents a systematic review aimed at assessing the prevalence of Epstein-Barr virus in patients with Burkitt's lymphoma. 135 studies published between 1969 and 202, which examined the prevalence of EBV in patients with BL were included in the study for meta-analysis. The authors present data from 4837 patients with BL lymphoma. The rationale for this type of research is not entirely clear, nor are the results. Is 57.5% the median value for the three clinical BL variants, or what does it reflect?

Three clinical variants of BL have been described based on the epidemiology of the cancer: endemic (eBL), sporadic (sBL) and immunodeficiency-associated (ID BL).

These variants are similar in morphology, immunophenotype and genetics. While sBL occurs outside of Africa and is rarely associated with EBV infection, eBL arises mainly in Africa and is associated with malaria endemicity and EBV infection.  The epidemiology for BL is different among various geographic areas. Differences in BL reflect distinct pathogenesis of the disease, and this well-known aspect could be valid. In addition, the other epidemiological factors and/or possible genetic predispositions still have an unclear role in the genesis of BL. To the present day, the exact cellular and molecular events that link EBV to increase the risk for BL remain to be elucidated.

EBV is detected in ~95% of EPD cases. EBV can also be found in about 30% of cases of sporadic Burkitt's lymphoma and in 25-40% of cases of immunodeficiency-associated Burkitt's lymphoma. To get the median, we sum up 95+30+40=165/3=55%.

Response: We would like to thank the academic editor and reviewer for taking their precious time to review this manuscript and give us comments. We actually tried to do a subgroup analysis splitting the three clinical variants; however, practically, we encountered a few issues, such as:

  1. Many of the included studies didn’t classify the BL into different variants.
  2. Many papers mention that we have three variants of BL, and after the samples were tested for EBV, the results were released for all BL without subdivision for variants.
  3. Many studies collected samples from different countries and institutes without mentioning the BL variants.

Meta-analysis doesn’t calculate the median, rather compiles all the study results and estimates the pooled estimate with a 95% confidence interval. Here, as we were unable to separate the three clinical variants, therefore, we probably found higher heterogeneity.

Reviewer 2 Report

the paper written by al-Kheisat is well written and flowing. Addresses the global prevalence of EBV in patients with BL. Introduction is well developeded. M&M are adequately described and results clair.

Author Response

Dear reviewer 2,

Thank you for your response. The manuscript has been spell checked, Thank you.

Reviewer 3 Report

This manuscript performed a meta -analysis of the prevalence of Epstein-Barr virus infection in patients with Burkitt lymphoma. The manuscript is well written, and it is easy to read and to interpret.

Heterogeneity was high. Nevertheless, the results are worth reporting.

Comment:

1) Line 46. Regarding infectious mononucleosis. Could you please add and/or change to "It is common among teenagers and young adults, especially college students."

2) Could you please make a Figure showing the different types of latency?

3) Line 57. Could you please add "Burkitt lymphoma (BL) is a highly aggressive B -cell non-Hodgkin lymphoma"

4) Line 63. Regarding the 3 subtypes of BL. Could you please add "They are histologically identical and have similar clinical behavior"

5) Could you please add in the introduction "BL is derived from germinal center B cells."

6) Line 92. MYC::IGH is mentioned. Nevertheless, please we aware that:

The Ig heavy chain gene on chromosome 14 (approximately 80 percent) – t(8;14)

The kappa light chain gene on chromosome 2 (approximately 15 percent) – t(2;8)

The lambda light chain gene on chromosome 22 (approximately 5 percent) – t(8;22)

7) Could you please add a histological description of the tumor (histopathological diagnosis)?

8) Would you agree with the Expression of CD21, the Epstein-Barr virus (EBV)/C3d receptor, depends on the EBV status of the tumors. Essentially, all cases of endemic BL are EBV positive and express CD21, whereas the vast majority of non-endemic BL in non-immunosuppressed patients are EBV negative and lack CD21 expression"?

9) Table 1. Regarding EBER+DLBCL. The distribution would be more "predominantly Asia"?

10) Line 114-115. The aim is to determine de prevalence of EBV virus in BL. But, using EBER evaluation in tissue samples? As shown in 108-110, there are many available techniques to identify EBV in human samples.

11) In line 193. I think it should be "<100".

12) Line 196. If you exclude cases of age >18. Won't you miss many cases of BL? Then, this study is a meta-analysis of pediatric BL? Of note, in sporadic BL: "The median age of the adult patients is 30 years, but there is also an incidence peak in elderly patients" ***Please note that manuscript 99 (Table 2) includes patients >18 years old.

13) How did you exclude "low-quality studies"? What is the criteria?

14) Lines 222-237. I would recommend moving this part to "appendix"

15) Lines 296-311 are very hard to read. Is there another way regarding the references of this paragraph?

16) Regarding Figure 1. I2 was 97%. Could you please comment on this: "The I2 statistic gives you an idea of the heterogeneity of the studies, i.e., how consistent they are. If the I2 value is >50%, it might mean the studies are inconsistent due to a reason other than chance. This might make the conclusions you draw from the forest plot questionable."? doi: 10.1177/21925682211003889

Did you use a fixed effect model or a random effect model?

17) Why not performing subgroups by BL -endemic, sporadic, and HIV? 17) Regarding Table 3.

Author Response

Reviewer 3

This manuscript performed a meta -analysis of the prevalence of Epstein-Barr virus infection in patients with Burkitt lymphoma. The manuscript is well written, and it is easy to read and to interpret.

Heterogeneity was high. Nevertheless, the results are worth reporting.

Response: We would like to thank the academic editor and reviewers for taking their precious time to review this manuscript and give us comments. We would like to explicitly state that we agree with all the comments as these helped us improve the quality of our paper. We have made a conscious effort to answer all the remarks in the paper as advised by the reviewers and highlighted changes with red colour in the revised manuscript for their convenience.

  1. Line 46. Regarding infectious mononucleosis. Could you please add and/or change to "It is common among teenagers and young adults, especially college students."

Response: Thank you for your insightful suggestions and comments; corrections have been carried out in the revised manuscript as per your comments.

Line 44-47

  1. Could you please make a Figure showing the different types of latency

Response: Thank you for your insightful suggestions and comments; corrections have been carried out in the revised manuscript through an adapted and designed table instead of the figure as per your comments.

Line 119-123

  1. Line 57. Could you please add "Burkitt lymphoma (BL) is a highly aggressive B -cell non-Hodgkin lymphoma"

Response: Thank you for your insightful suggestions and comments; corrections have been carried out in the revised manuscript as per your comments.

Line 60-61

  1. Line 63. Regarding the 3 subtypes of BL. Could you please add "They are histologically identical and have similar clinical behavior"

Response: Thank you for your insightful suggestions and comments; corrections have been carried out in the revised manuscript as per your comments.

Line 79-80

  1. Could you please add in the introduction "BL is derived from germinal center B cells."

Response: Thank you for your insightful suggestions and comments; corrections have been carried out in the revised manuscript as per your comments.

Line 63

  1. Line 92. MYC::IGH is mentioned. Nevertheless, please we aware that:

The Ig heavy chain gene on chromosome 14 (approximately 80 percent) – t(8;14)

The kappa light chain gene on chromosome 2 (approximately 15 percent) – t(2;8)

The lambda light chain gene on chromosome 22 (approximately 5 percent) – t(8;22)

Response: Thank you for your insightful suggestions and comments; corrections have been carried out in the revised manuscript as per your comments.

Line 127-131

  1. Could you please add a histological description of the tumor (histopathological diagnosis)?

Response: Thank you for your insightful suggestions and comments; corrections have been carried out in the revised manuscript as per your comments.

Line 64-69

  1. Would you agree with the Expression of CD21, the Epstein-Barr virus (EBV)/C3d receptor, depends on the EBV status of the tumors. Essentially, all cases of endemic BL are EBV positive and express CD21, whereas the vast majority of non-endemic BL in non-immunosuppressed patients are EBV negative and lack CD21 expression"?

Response: Thank you for your insightful suggestions and comments; Yes, I agree. The corrections have been carried out in the revised manuscript as per your comments.

Line 69-73

  1. Table 1. Regarding EBER+DLBCL. The distribution would be more "predominantly Asia"?

Response: Yes, correct, the predominance of EBV+ DLBCL in Asia is more than Europe. However, the incidence of EBV-positive DLBCL is less than 15%.

Could you please follow up the following references.

  1. Beltran, B.E.; Castillo, J.J.; Morales, D.; de Mendoza, F.H.; Quinones, P.; Miranda, R.N.; Gallo, A.; Lopez-Ilasaca, M.; Butera, J.N.; Sotomayor, E.M. EBV-positive diffuse large B-cell lymphoma of the elderly: a case series from Peru. Am J Hematol 2011, 86, 663-667, doi:10.1002/ajh.22078.
  2. Beltran, B.E.; Castro, D.; Paredes, S.; Miranda, R.N.; Castillo, J.J. EBV-positive diffuse large B-cell lymphoma, not otherwise specified: 2020 update on diagnosis, risk-stratification and management. Am J Hematol 2020, 95, 435-445, doi:10.1002/ajh.25760.
  3. Hwang, J.; Suh, C.H.; Won Kim, K.; Kim, H.S.; Armand, P.; Huang, R.Y.; Guenette, J.P. The Incidence of Epstein-Barr Virus-Positive Diffuse Large B-Cell Lymphoma: A Systematic Review and Meta-Analysis. Cancers (Basel) 2021, 13, 1785, doi:10.3390/cancers13081785.
  4. Line 114-115. The aim is to determine de prevalence of EBV virus in BL. But, using EBER evaluation in tissue samples? As shown in 108-110, there are many available techniques to identify EBV in human samples.

Response: The included studies have a long-time interval running from 1969 to 2020; during this time, many invented methods were used to assess the prevalence of EBV in patients with BL. As a result, to determine the overall prevalence of EBV in patients with BL, we included all methods in this review. 

When you are back to Table 3., you will find the sample types are different accordingly to the method of detection.

  1. In line 193. I think it should be "<100".

Response: Thank you for your insightful suggestions and comments; Yes, correct. The corrections have been carried out in the revised manuscript as per your comments.

Line 230

  1. Line 196. If you exclude cases of age >18. Won't you miss many cases of BL? Then, this study is a meta-analysis of pediatric BL? Of note, in sporadic BL: "The median age of the adult patients is 30 years, but there is also an incidence peak in elderly patients" ***Please note that manuscript 99 (Table 2) includes patients >18 years old.

Response: We do here subgroup analysis as we have many papers (44 included papers) with ages less than 18 only without any age other categories (i.e., above 18).

We can't say that this study is a meta-analysis of paediatric BL for some reasons:

  1. The included studies didn't have one age category (i.e., less than 18).
  2. We don't have enough studies with only ages older than 18 years old. As we have only five studies, the total number of patients included is around 25 corresponding to 2187 patients in ages less than 18. Therefore, we didn't proceed with subgroup analysis.
  3. The included studies have different age categories (≤18, 45<Age>18, and ≥45 ), with about 27 papers included.

  1. How did you exclude "low-quality studies"? What is the criteria?

Response: The included studies were assessed using Joanna Briggs Institute critical appraisal tools. The studies were defined as poor quality (high risk of bias), moderate quality (moderate risk of bias), or high quality (low risk of bias) if the overall score was ≤49%, 50–69%, or ≥70, respectively.

Each type of study (Case-control, Cohort, and Cross-sectional) have their own checklist to assess the methodological quality of a study and to determine the extent to which a study has addressed the possibility of bias in its design, conduct and analysis.

Could you please follow the following reference, and Tables S2, S3, and S4 for quality assessment questions.

  1. Seak, Y.S.; Nor, J.; Tuan Kamauzaman, T.H.; Arithra, A.; Islam, M.A. Efficacy and Safety of Intranasal Ketamine for Acute Pain Management in the Emergency Setting: A Systematic Review and Meta-Analysis. J Clin Med 2021, 10, 3978, doi:10.3390/jcm10173978.

For calculation,

  • Paper high quality (i.e., ≥70) =

(68 [cross-sectional] + 16 [case-control] + 4 [cohort])/135 *100%= 65.2%

  • Paper moderate quality (i.e., 50–69%) =

(11 [cross-sectional] + 20 [case-control] + 9 [cohort])/135 *100%= 29.6%

  • Paper low quality (i.e., ≤49%,) =

(0 [cross-sectional] + 3 [case-control] + 5 [cohort])/135 *100%= 5.9%

  1. Lines 222-237. I would recommend moving this part to "appendix"

Response: Thank you for your insightful suggestions and comments. These rows show subgroup categories with their related papers. If moved to the appendix part, they will disrupt reference sequences, and this subgroup heading is considered an important part of describing study characteristics.

  1. Lines 296-311 are very hard to read. Is there another way regarding the references of this paragraph?

Response: Unfortunately, there is no other way to write in a different manner. However, the references in the after-published paper appear in blue, which may make them easier to read.

  1. Regarding Figure 1. I2 was 97%. Could you please comment on this: "The I2 statistic gives you an idea of the heterogeneity of the studies, i.e., how consistent they are. If the I2 value is >50%, it might mean the studies are inconsistent due to a reason other than chance. This might make the conclusions you draw from the forest plot questionable."? doi: 10.1177/21925682211003889

Did you use a fixed effect model or a random effect model?

Response: Thank you for your insightful suggestions and comments. In incidence or prevalence meta-analysis, high heterogeneity is a very common phenomenon. Unfortunately, there are no guidelines or thresholds for substantial heterogeneity yet. The concept of heterogeneity having >50% or <50% comes from a randomized controlled trial (RCT) or other observational studies estimating OR/RR. Let us share some prevalence-based meta-analysis results and their heterogeneities.

  1. Adil, S.O.; Islam, M.A.; Musa, K.I.; Shafique, K. Prevalence of Metabolic Syndrome among Apparently Healthy Adult Population in Pakistan: A Systematic Review and Meta-Analysis. Healthcare202311, 531. https://doi.org/10.3390/healthcare11040531
  • Heterogeneities = 100%
  1. Kundu, S.; Alam, S.S.; Mia, M.A.-T.; Hossan, T.; Hider, P.; Khalil, M.I.; Musa, K.I.; Islam, M.A. Prevalence of Anemia among Children and Adolescents of Bangladesh: A Systematic Review and Meta-Analysis.  J. Environ. Res. Public Health202320, 1786. https://doi.org/10.3390/ijerph20031786
  • Heterogeneities = 97%
  1. Ahmad, M.A.; Ab Rahman, S.; Islam, M.A. Prevalence and Risk of Infection in Patients with Diabetes following Primary Total Knee Arthroplasty: A Global Systematic Review and Meta-Analysis of 120,754 Knees.  Clin. Med.202211, 3752. https://doi.org/10.3390/jcm11133752
  • Heterogeneities = 87-92%
  1. Hajissa, K.; Islam, M.A.; Sanyang, A.M.; Mohamed, Z. Prevalence of intestinal protozoan parasites among school children in africa: A systematic review and meta-analysis. PLOS Neglected Tropical Diseases 2022, 16, e0009971, doi:10.1371/journal.pntd.0009971.
  • Heterogeneities = 100%

Did you use a fixed effect model or a random effect model?

The mode used here is random effect model.

  1. Why not performing subgroups by BL -endemic, sporadic, and HIV? 17) Regarding Table 3.

Response: Thank you for your insightful suggestions and comments. It’s difficult to do, as many of the included studies didn’t classify the BL into different categories. Furthermore, some studies merge samples from different regions without mentioning anything about the type of BL.

Reviewer 4 Report

The review by Al-Khreisat et al. nicely summarizes available data on the prevalence of the Epstein-Barr virus (EBV) in Burkitt’s lymphoma (BL). I have no major concerns with this review which is very thorough and well written.

There a few points that should be addressed:

1.      Lines 87-90. The six EBNAs as well as the LMPs etc. are only expressed in replicating cells and in tumour cells in post-transplant lymphoproliferative disease under immunosuppression as well as in replicating B-cells. In BL, only non-coding RNAs (EBERs, miRNAs) and EBNA1 are expressed while EBV-positive Hodgkin’s lymphoma (HD) and nasopharyngeal carcinoma (NPC) additionally express LMPs. It would help the reader if a Table with the various forms of latency including type O latency (in resting memory B-cells) would be included.

2.      In the introduction, it should be mentioned that increasing evidence links an infection with EBV to multiple sclerosis.

3.      Table 1: undifferentiated nasopharyngeal carcinoma like ENKTL are always EBV positive while differentiated NPC are mostly negative. This should be outlined in Table 1.

4.      Table 3: “Subgroups”: Number of BL patients have EBV: should better read “positive for EBV”

5.      Discussion: it would be nice if the authors would comment on “publication bias for prevalence of EBV virus in BL (p=0.0034).” (line 288).

6.       Also, the sentence in line 288 should read …EBV in BL….

Author Response

Reviewer 4

The review by Al-Khreisat et al. nicely summarizes available data on the prevalence of the Epstein-Barr virus (EBV) in Burkitt’s lymphoma (BL). I have no major concerns with this review which is very thorough and well written.

There a few points that should be addressed:

Response: We would like to thank the academic editor and reviewers for taking their precious time to review this manuscript and give us comments. We would like to explicitly state that we agree with all the comments as these helped us improve the quality of our paper. We have made a conscious effort to answer all the remarks in the paper as advised by the reviewers and highlighted changes with red colour in the revised manuscript for their convenience.

  1. Lines 87-90. The six EBNAs as well as the LMPs etc. are only expressed in replicating cells and in tumour cells in post-transplant lymphoproliferative disease under immunosuppression as well as in replicating B-cells. In BL, only non-coding RNAs (EBERs, miRNAs) and EBNA1 are expressed while EBV-positive Hodgkin’s lymphoma (HD) and nasopharyngeal carcinoma (NPC) additionally express LMPs. It would help the reader if a Table with the various forms of latency including type O latency (in resting memory B-cells) would be included.

Response: Thank you for your insightful suggestions and comments; corrections have been carried out in the revised manuscript as per your comments.

Line 119-123

  1. In the introduction, it should be mentioned that increasing evidence links an infection with EBV to multiple sclerosis.

Response: Thank you for your insightful suggestions and comments; corrections have been carried out in the revised manuscript as per your comments.

Line 99-107

  1. Table 1: undifferentiated nasopharyngeal carcinoma like ENKTL are always EBV positive while differentiated NPC are mostly negative. This should be outlined in Table 1.

Response: Thank you for your insightful suggestions and comments; corrections have been carried out in the revised manuscript as per your comments for undifferentiated nasopharyngeal carcinoma; while for differentiated NPC, could you please provide a reference?

Line 116, Table 1

  1. Table 3: “Subgroups”: Number of BL patients have EBV: should better read “positive for EBV”

Response: Thank you for your insightful suggestions and comments; corrections have been carried out in the revised manuscript as per your comments.

Line 319

  1. Discussion: it would be nice if the authors would comment on “publication bias for prevalence of EBV virus in BL (p=0.0034).” (line 288).

Response: Thank you for your insightful suggestions and comments; corrections have been carried out in the revised manuscript as per your comments.

Line 390-395

  1. Also, the sentence in line 288 should read …EBV in BL…

Response: Thank you for your insightful suggestions and comments; corrections have been carried out in the revised manuscript as per your comments.

Line 329

Round 2

Reviewer 1 Report

The authors generally responded to the comments of the reviewers and explained the inability to assess the prevalence of EBV for the three clinical variants. However, the manuscript still contains serious implications:

The fact that EBV is very common in the general population, and  only a very small fraction of infected individuals develop EBV-associated pathologies, indicates that other risk factors such as immune system status, genetic variability/predisposition and environmental factors are also can contribute in the development of EBV-associated pathologies.

1.     Thus, country’s social demographic index (SDI) should be applied for the study.

The authors sought to estimate the worldwide prevalence of Epstein-Barr virus in patients with Burkitt's lymphoma, but they provide many non-BL data on EBV-associated diseases (Table 1 and Table 2).

2.     How these data are used to estimate the prevalence of EBV in patients with BL is not clear, and they cannot be used to achieve the main goal of the study.

3.     It would be nice to present the number of patients included in the study based on 135 publications.

4.     The authors suggest EBV testing as an alternative for predicting and evaluating the clinical status of BL, but do not propose a method for use.

Memory B cells infected with EBV may be present in healthy people in the latent stage for a long time in the blood. It has been estimated that there is 1–50 copies of EBV-DNA/106 WBCs in healthy carrier’s blood, although EBV-DNA is undetectable in serum or plasma.

Artificial intelligence can be used for research.

Author Response

Reviewer 1

Comment: The authors generally responded to the comments of the reviewers and explained the inability to assess the prevalence of EBV for the three clinical variants. However, the manuscript still contains serious implications:

Response: We would like to thank the academic editor and reviewer for taking their precious time to review this manuscript and give us comments. As we mentioned before, we actually tried to do a subgroup analysis splitting the three clinical variants; however, practically, we encountered a few issues, such as:

  1. Many of the included studies didn’t classify the BL into different variants.
  2. Many papers mention that we have three variants of BL, and after the samples were tested for EBV, the results were released for all BL without subdivision for variants.
  3. Many studies collected samples from different countries and institutes without mentioning the BL variants.
  4. In Almost all the included studies, patients in one study come from a single country.

We hope this clarifies the reason why we were unable to conduct that desired subgroup analysis. Thank you.

Comment: The fact that EBV is very common in the general population, and only a very small fraction of infected individuals develop EBV-associated pathologies, indicates that other risk factors such as immune system status, genetic variability/predisposition and environmental factors are also can contribute in the development of EBV-associated pathologies.

Response: Thank you for your insightful suggestion and comment; we have incorporated this in our updated manuscript; hopefully, this time, this part reads better (Lines 58–61).

Comment: Thus, country’s social demographic index (SDI) should be applied for the study.

Response: Thank you for your insightful suggestion and comment; in the revised manuscript, we have introduced a new subgroup analysis based on the SDIs of the included studies and corresponding countries. We hope that this part is alright now. Based on the new subgroup analysis (Table 4, Supplementary Figure S1: T-X), the following lines have been added:

Line 245-251

Line 301-307

Line 351-358

Table 3 (360)

Line 449-454

Line 461-462

Line 471

Supplementary Figure S1

Comment: The authors sought to estimate the worldwide prevalence of Epstein-Barr virus in patients with Burkitt's lymphoma, but they provide many non-BL data on EBV-associated diseases (Table 1 and Table 2). How these data are used to estimate the prevalence of EBV in patients with BL is not clear, and they cannot be used to achieve the main goal of the study.

Response: Many thanks for indicating the issue.

Previously, Table 1 was added to present the incidences and other EBV-associated diseases published previously. However, after considering the comment of the reviewer, we also feel the same, and therefore, we have decided to remove this avoidable table. Thank you.

In the case of Table 2, it was added in response to the other two reviewers’ suggestions (Reviewer 3 and Reviewer 4) previously, and this table provides an overview of latency programs of EBV-associated malignancies along with expressed genes and products that show which EBV-latency program is associated with BL. As this table is not majorly influencing the integrity of this systematic review, we would like to keep this insightful table. We hope that it’s alright. Thank you.

Comment: It would be nice to present the number of patients included in the study based on 135 publications.

Response: Thank you for your suggestion; however, we are afraid that, as these data have already been presented in all the forest plots, to avoid duplication, we would like to avoid adding identical data in figures. We hope it’s understandable. Thank you.

Comment: The authors suggest EBV testing as an alternative for predicting and evaluating the clinical status of BL, but do not propose a method for use.

Response: Thank you for your wonderful suggestion; the proposed method for use has been added in the revised manuscript (Lines 448 and 449). We hope it’s alright now. Thank you.

Comment: Memory B cells infected with EBV may be present in healthy people in the latent stage for a long time in the blood. It has been estimated that there is 1–50 copies of EBV-DNA/106 WBCs in healthy carrier’s blood, although EBV-DNA is undetectable in serum or plasma. Artificial intelligence can be used for research.

Response: Thank you for your insightful suggestion. In the revised manuscript, we have added the required information between lines 153 and 164. We hope that this part reads better now. Thank you.

Round 3

Reviewer 1 Report

In the revised version of the manuscript, the authors responded to the reviewer's comments and explained the issues raised, clarifying them in the text.